# Long Non-Coding RNAs: Tools for Understanding and Targeting Cancer Pathways

**DOI:** 10.3390/cancers14194760

**Published:** 2022-09-29

**Authors:** Gaurav Kumar Pandey, Chandrasekhar Kanduri

**Affiliations:** 1Department of Zoology, Banaras Hindu University, Varanasi 221005, India; 2Department of Medical Biochemistry and Cell Biology, The Sahlgrenska Academy, Institute of Biomedicine, University of Gothenburg, SE-40530 Gothenburg, Sweden

**Keywords:** long non-coding RNA, cancer, non-coding RNA, oncogenes, tumor suppressors

## Abstract

**Simple Summary:**

Long non-coding RNA (lncRNA) is a type of RNA molecule that show striking resemblance to mRNA in terms of synthesis and structure but lacks protein coding capacity. High-throughput RNA-sequencing technologies revealed that they are present in tens of thousands, outnumbering the protein coding genes. Functional investigations over the last two decades reveal that they show highly tissue-, cell type-, and developmental-specific expression and regulate biological processes that control differentiation and development. More importantly, recent evidence suggests that lncRNA are highly dysregulated in cancers and play a crucial role in cancer development and progression through regulating oncogenic and tumor suppressor pathways, cellular metabolism, and the immune response and tumor microenvironment. This article reviews the technologies to identify oncogenesis related lncRNAs and the efforts to understand their functional role in tumor development. We also pay attention towards whether they can serve as potential therapeutic candidates in our fight against cancer.

**Abstract:**

The regulatory nature of long non-coding RNAs (lncRNAs) has been well established in various processes of cellular growth, development, and differentiation. Therefore, it is vital to examine their contribution to cancer development. There are ample examples of lncRNAs whose cellular levels are significantly associated with clinical outcomes. However, whether these non-coding molecules can work as either key drivers or barriers to cancer development remains unknown. The current review aims to discuss some well-characterised lncRNAs in the process of oncogenesis and extrapolate the extent of their decisive contribution to tumour development. We ask if these lncRNAs can independently initiate neoplastic lesions or they always need the modulation of well characterized oncogenes or tumour suppressors to exert their functional properties. Finally, we discuss the emerging genetic approaches and appropriate animal and humanised models that can significantly contribute to the functional dissection of lncRNAs in cancer development and progression.

## 1. Introduction

A surprising finding of high-throughput transcriptomic analyses in early 2000 gave us an interesting fact that transcription is not just restricted to the genome occupied by protein-coding genes, but more than two-thirds of the genome is dynamically transcribed, giving rise to tens of thousands of long non-coding RNAs (lncRNAs), thus adding further complexity in uncovering the functional properties of the genome. These high-throughput investigations also revealed that lncRNA display a high level of cell and tissue-specific expression. Functional studies over the last two decades showed that they take part in diverse biological function by regulating gene expression at transcriptional, post-transcriptional, and post-translational levels [1,2,3]. Thus, it is no surprise that the aberrant expression of lncRNAs has been linked to cancer. 

These efforts are reflected in almost two decades of research investigating the role of altered lncRNAs in the origin and progression of cancer. Most studies have demonstrated that the expression of several lncRNAs is significantly associated with disease progression, aggressiveness, or clinical outcomes [4,5]. Importantly, functional and mechanistic studies in cell and animal models of cancers have revealed that the levels of specific lncRNAs can influence the components of the crucial cancer pathways such as cell cycle regulation, cellular metabolism, immune response, epigenetic regulation, etc. [6,7]. As a result, there is a lot of academic and clinical interest in identifying novel cancer-associated transcripts along with the further characterisation of known lncRNA candidates. However, the biggest roadblock to utilising the full clinical and therapeutic potential of these non-coding species is that we still do not know the functional role of many cancer-altered lncRNAs. Recent evolution in technologies such as CRISPR-based genetic screens has enabled researchers to functionally annotate lncRNAs and catalogue them in dedicated databases. These recent efforts show that recurring molecular mechanisms for lncRNA are now realised in regulating the processes strongly connected to cancer development and progression.

In the current review, we will elaborate on our present understanding of the functional role of lncRNAs in cancer pathogenesis. Further, we will discuss the available strategies for lncRNA investigation and how they can influence various dimensions of tumour progression and development. Additionally, we attempt to critically analyse the significant hurdles and new possibilities of moving lncRNAs beyond just markers of cancer progression to realistic therapeutic targets. Finally, we will discuss if lncRNAs are just hype or reality in cancer research, their overall significance, and future directions. Overall, the current review focuses on lncRNAs that have been functionally well characterized to emphasize the need to uncover the molecular mechanisms behind the dysregulation of lncRNAs in the process of carcinogenesis. 

## 2. General Features and Roles of Long Non-Coding RNAs

LncRNAs are non-protein-coding RNA species that are around 200 base pair or more in size. The recent annotations estimate their numbers to be around 31,678 in humans but this number may exceed [8,9,10,11]. However, how many of these lncRNAs are functional is still not known. Compared to protein coding genes, lncRNAs display highly lineage-restricted expression and show poor conservation at the sequence level. On the contrary, lncRNA promoters show a high level of conservation. However, emerging evidence indicates that the primary conservation in lncRNAs is more intricate and should not be visualized only from the sequence perspective. LncRNAs can be conserved at multiple levels such as secondary or higher-order tertiary structure, binding motifs for interacting proteins, functional features, and genomic location [12,13]. In addition to structural complexity, lncRNAs adopt diverse molecular mechanisms for executing their regulatory functions. They can act as scaffolds for targeting chromatin modifying enzymes, transcriptional factors, and regulators of splicing, and as sponges or decoys for microRNAs and transcription factors, etc. Details of molecular mechanisms of gene regulation by lncRNAs have been discussed in greater detail in a number of reviews [1,2,14].

## 3. Understanding the Contribution of lncRNA in Cancer Development

Tumours, in general, are comprised of heterogeneous cells with varying cell states. A combination of alterations in genetic, signalling, metabolic, and microenvironments converge to dictate the fate of tumour progression. According to the protein-centric view of cancer development, mutations/alterations in key genes such as *EGFR*, *ALK*, *BRAF*, *KRAS, TP53*, and *MYC* mainly drive the oncogenic events and thus influence the possible clinical outcomes [15,16,17,18,19]. However, this information is insufficient as the cancer treatments based on protein-coding genes provide a limited cure, especially for relapsed cancer patients. As the non-coding genome has been gaining traction lately among researchers due to its surprising functions, it is pertinent to examine its contribution to cancer development. In fact, it is vital to analyse if lncRNAs can also work as master regulators of tumorigenesis or if they are just followers of protein-coding master regulators. In the current section, we will examine this statement in light of well-known lncRNAs reported in more than one cancer and associated with key aspects of cancer development (Figure 1). 

### 3.1. LncRNA in Oncogenic Pathways

LncRNAs can act as promoters of key oncogenic events associated with cancer development. These events include cell growth, invasion, metastasis, suppression of cell death pathway, alteration in DNA damage response, etc. *HOTAIR* is one such lncRNA reported in oncogenic roles across multiple cancer types. Moreover, it can act as an independent predictor of clinical outcome and thus is suggested as an ideal biomarker for poor prognosis. It is known to regulate its downstream effectors via targeting well-known repressive chromatin modifiers, Polycomb Repressive Complex 2 (PRC2), and LSD1 complex (H3K4 demethylase) to its target promoters [20]. This *HOTAIR*-induced reprogrammed epigenome, in turn, promotes the gene expression pattern required for breast cancer metastasis. Additionally, it can function as a competitive endogenous RNA (ceRNA) and regulate the critical micro-RNA dependent TGF-β, PI3K-AKT, Wnt/β, and VEGF oncogenic-signalling cascades [21,22]. 

Another oncogenic lncRNA is the leukemia-induced non-coding activating RNA 1 (*LUNAR1*), initially identified as a critical downstream target of oncogenic Notch1 in T cell acute lymphoblastic leukaemia (T-ALL) [23]. It executes its oncogenic regulatory function by controlling the activation of insulin-like growth receptor 1 (IGF1R), a Notch1 target that mediates growth/survival signals in T-ALL. *LUNAR1* functions as an enhancer-like lncRNA whereby it recruits mediator complex to the intronic *IGF1R* enhancer and thus fully activates *IGF1R* promoter via DNA looping. As oncogenic Notch1 and over-activation of IGF1 are known to promote cancer development, *LUNAR1* may be active in other cancers with Notch1 alterations. Indeed, its elevated expression was found to be associated with colorectal cancer aggressiveness, and thus this lncRNA may work as a promising prognostic marker as well as a therapeutic target for tumours driven by altered Notch1 signalling. Another lncRNA, *NALT*, is also reported as a potential activator of NOTCH1 [24]. Other pro-oncogenic lncRNAs, such as *DANCR* and *AK023948* are known to function as positive regulators of AKT pathway [25,26].

Recent studies have turned to cancer cell hallmark-based screens to identify tumor driving lncRNAs. For example, deregulated cell cycle, one of the well-studied cancer cell hallmarks, has been used to identify lncRNA-based oncogenic drivers. Using an S-phase based nascent capture assay, a recent investigation has identified 570 S-phase enriched lncRNAs that show significant differential expression in at least one tumor type across pan-cancer TCGA data sets. Mechanistic investigations on one of the top S-phase enriched cancer associated lncRNAs *SCAT7* (*ELF3-AS1*) revealed that it promotes tumorigenesis through activating fibroblast growth factor (FGF/FGFR)-dependent PI3K/AKT and MAPK pro-survival pathways via interacting with hnRNPK/YBX1 complex [4]. Besides, it maintains genome integrity through regulating Topoisomerase 1 turnover [27]. Thus, S-phase enriched cancer associated lncRNAs will serve as an interesting resource for identifying potential new targets for cancer treatment.

Another intelligent approach that has yielded identifications of lncRNAs in oncogenic roles has been the genomic and transcriptomic analysis of somatic copy number alterations (SCNAs), frequently observed across tumour types. SCNA analysis revealed two interesting lncRNAs, *FAL-1* and *PLANE,* that, like their protein-coding counterparts at amplified regions (such as *MYCN* gene amplification in neuroblastoma tumors), have been functionally characterized for their tumour-promoting roles [28,29]. *FAL-1* maps to the frequently amplified region on chromosome 1, and its dysregulation is observed across multiple tumour types. Moreover, its association with poor prognosis makes it an attractive candidate for disease prognosis and targeted therapy. *FAL-1* exerts its oncogenic functions, like promoting cell proliferation and metastasis via diverse regulatory mechanisms such as stabilising epigenetic repressor BMI 1, enhancer-like functions, phosphorylation of STAT3, and upregulation of EMT proteins. Similar to *FAL-1*, *PLANE* maps to the chromosome 3q region and promotes oncogenic pathways by regulating an alternative splicing programme [29]. 

Apart from their more direct function, lncRNAs can also work either as modulators or downstream effectors of oncogenic drivers. This is best seen in the case of the known transcription factor and oncogenic driver, *MYC*. LncRNAs such as *PVT1*, *PCAT1*, *CCAT1,* and *CCAT2* affect the expression of *MYC*, where *PVT1* post-transcriptionally stabilizes MYC, *PCAT1* functions as a ceRNA preventing the downregulation of MYC by miR-34a [30,31]. *CCAT1* and *CCAT2* are the two enhancer-driven lncRNAs located upstream of the *MYC* gene that influences its expression via cis-regulating mechanisms [32,33]. In addition to these upstream regulators of *MYC*, 19 lncRNAs as downstream effectors of *MYC* have been reported [34].

All these examples clearly indicate how oncogenic lncRNAs are instrumental in the process of carcinogenesis. By functioning as modulators of epigenetic machinery, signalling pathways, long-range chromatin interactions, oncogenes, and micro-RNA functions, lncRNAs seem to play a significant role in the oncogenic process. Due to their elevated expression and association with unfavourable clinical outcomes, oncogenic lncRNAs are ideal candidates for prognostic markers and make attractive targets for therapeutic interventions. Although some of the lncRNAs, such as *HOTAIR*, have been extensively investigated as oncogenic drivers in multiple cancer types, we are still far from understanding the major common oncogenic pathways by which they induce neoplastic transformation. Thus, more work is needed on these lines to develop lncRNAs as more reliable targets for cancer therapy.

### 3.2. LncRNAs in Tumor Suppressor Pathways

In addition to their role in promoting tumour development, lncRNAs are also known to be involved in tumour suppressive roles. We will discuss some well-established lncRNAs which are exclusively known to function as suppressors of tumour development across many cancers. Further, we will extrapolate these findings in speculating whether lncRNA work as bonafide tumour suppressors or just as assistants to protein-coding TSGs (tumour suppressor genes). 

Well-studied examples of tumour suppressor lncRNAs are known in Neuroblastoma (NB), an embryonal childhood cancer of peripheral nervous system. Initial genome-wide association studies (GWAS) in these tumours uncovered a risk locus 6p22.3 with clusters of risk-associated single nucleotide polymorphisms (SNPs). What is more fascinating is that these SNPs are present at a genomic location with no protein-coding genes. These genomic hot spots consist of sense and antisense pairs of lncRNAs *CAS15* and *NBAT1,* whose reduced levels are observed in high-risk NBs and are associated with poor prognosis [35,36]. The lower expression of sense *CASC-15* and antisense *NBAT1* in high-risk NBs is caused due to epigenetic suppression of their promoter. *NBAT1* exerts its tumour suppressive activity by epigenetic silencing of genes such as *SOX9*, *VCAN*, and *OSMR* via its association with chromatin repressor, EZH2 [37,38]. It also promotes neuronal differentiation by modulating neuronal-specific transcription factor NRSF/REST. *NBAT1*’s sense counterpart *CASC15* has several isoforms and among which *CASC15-003* and *CASC15-004* have been shown to display tumor suppressor functions. Of note, the isoform *CASC15-003* shares common biological pathways with *NBAT1* in promoting tumor suppression and inducing neuronal differentiation. For example, both the sense (*CASC15-003*) and antisense (*NBAT1*) lncRNAs regulate the expression of SOX9, a well-characterized neural crest-specific transcription factor, through controlling CHD7 stability by modulating the nucleolar-specific localization of deubiquitinating enzyme USP36 by interfering with USP36 and NPM1 interaction. Consistent with these observations, *NBAT1* and *CASC15-003* rescue each other’s functions in retinoic acid induced neuronal differentiation, thus these lncRNAs are functionally complement each in promoting their functions. These data also indicate that *NBAT1* controls SOX9 expression at the transcriptional and post-transcriptional level, whereas *CASC15* regulates SOX9 expression at the transcriptional level. Thus, SOX9 appears to be one of the key genes in neuroblastoma progression, whose oncogenic functions are controlled by *NBAT1* and *CASC15-003*. Their optimal levels are essential to ensure normal neuronal differentiation and suppressing cancer progression and development. The broader clinical significance of these lncRNAs is reflected by their identification as prognostic markers in other cancers. For instance, *NBAT1* is a potential tumor suppressor, and its higher expression is associated with good clinical outcomes in cancers such as NSCLC, oesophageal cancer, HCC, renal cell carcinoma, and breast cancer [39,40,41,42]. Thus, *NBAT1* is a bona fide TSG.

*NORAD* (non-coding RNA activated by DNA damage) is another candidate lncRNA with tumour-suppressive properties, initially identified in DNA damage screen in colon cancer cell lines. It is a conserved, cytoplasmic, localised, abundant RNA with multiple repeat sequences believed to be assembly platforms of Pumilio ribonuclear protein complexes (PUM1 and PUM2). The lncRNA *NORAD* prevents aberrant mitosis by inhibiting Pumilio (PUM) proteins [43]. Its TSG functions mainly occur as a decoy for PUM1 and PUM2 by nucleating the formation of phase-separated Pumilio condensates, termed NP bodies [44]. Disruption of NORAD-mediated PUM phase separation leads to hyperactivation of Pumulio proteins and genome instability. Similarly, it is also reported to sequester pro-metastatic regulator, S100-P, thus suppressing metastasis in breast and lung cancers [45]. 

*GAS5 (Growth Arrest-Specific Transcript 5)* is also an interesting tumour suppressor lncRNA whose lowered expression is associated with poorer clinical outcomes in a variety of human malignancies [46]. It has been functionally implicated in wide cancer-related processes such as cell proliferation, invasion, metastasis, epithelial to mesenchymal transition (EMT), drug resistance, etc. [47]. This plethora of functions are carried out using diverse mechanisms such as association with chromatin-modifying enzymes, acting as a sponge for mi-RNAs and formation of DNA-RNA triplex structures.

Like *GAS5*, *MEG3*, an imprinted lncRNA with maternal-specific expression, has been shown to act as a tumor suppressor in several cancers by being in strong functional connection with epigenetic machinery. Functional investigations have shown that this lncRNA suppresses tumors through downregulating oncogenic pathways such as MYC and TGF-β and activating p53 and RB. Published data indicates that *MEG-3*-dependent suppression of oncogenic pathways involves large-scale epigenetic programming. *MEG3* has been shown to have distinct interactions with the PRC2 complex and its cofactor, JARID2. Such interactions have been implicated in the regulation of expression of transforming growth factor-β (TGF-β) pathway genes. Further mechanistic investigations revealed that *MEG-3* suppresses TGF-β pathway genes through forming triplexes at purine-rich GA stretches to recruit chromatin modifiers [48].

LncRNA molecules also exert their tumour-suppressive functions by either inhibiting cancer promoting oncogenic pathways or fine-tuning the response of protein coding TSGs. *NKILA* (NF-κB interacting LncRNA) is one such tumour-suppressive RNA that works as a feedback regulator of NF-κB activity [49]. Its decreased expression is associated with poor outcomes in patients. *NKILA* acts as a scaffold to recruit NF-κB/IκB complex and prevents phosphorylation of IκB by IKK. This RNA is upregulated by NF-κB and forms a negative feedback loop of NF-κB. Additionally, a mouse lncRNA, *Lethe*, is induced by NF-κB and functions as a negative feedback regulator of NF-κB [50]. However, the significance of *Lethe* in cancer remains unknown. LncRNAs’ function as fine-tuners of TSG activity is best exemplified in the case of the well-known TSG, *TP53* [51]. Several lncRNAs are reported as components of a p53-mediated network where they work either as its modulators or downstream effectors. Decreased expression of maternally expressed genes 3 (*MEG3*) lncRNA is seen in many cancers, and it is known to activate p53 and thus modulate its downstream target activity [52]. *MEG-3* functional connection with p53 has been well established by several studies. A recent work has characterized conserved pseudoknots in *MEG3* that are responsible for the p53 stimulation, revealing the structural basis for the p53 stimulation [53]. Although *MEG-3* has been shown to directly interact with p53 protein, several studies have implicated the indirect role of *MEG-3* in p53 pathway stimulation [54]. Thus, functional dissection of this well-connected tumor suppressor pathway will help in uncovering the molecular basis of MEG-3-p53-dependent tumor suppression. Another lncRNA induced by DNA damage is conserved lncRNA, *DINO* (damage induced non-coding), present upstream of *CDKN1A* [55]. In response to DNA damage, it binds and stabilizes p53 and thus activates the p53-mediated gene network. Its functional relevance is elucidated in the transgenic mice (knockout or inactivated promoter) of *DINO* showing dampened p53-induced DNA damage response. Along similar lines, *NBAT1* is a p53-responsive lncRNA, that in turn regulates p53 subcellular levels through controlling p53 nuclear export machinery CRM1. Higher *NBAT1* expression correlates with increased p53 signaling, and depletion of *NBAT1* altered CRM1 function, contributing to the loss of p53-dependent nuclear gene expression [56]. Based on NBAT1-CRM1-p53 functional connection, Nutlin-3a and Selinexor treatment has been proposed for the treatment of Neuroblastoma patients. On the other hand, lncRNAs such as *lincRNA-p21*, Lincprint, *Tug1,* and *PANDA* form the downstream components of the p53 network where *lincRNA-p21* and *Lincprint* exert their p53 tumor suppressive function via association with hnRNPK and PRC2, respectively. *PANDA*, p53-inducible lncRNA, sequesters and regulate NFYA (nuclear transcription factor Y, alpha) levels to control the expression of pro-survival and pro-apoptotic genes [51]. 

### 3.3. Context-Specific Roles of LncRNAs as Oncogenes or TSGs

It is evident from the differential expression analyses of lncRNAs in pan-cancer TCGA data analyses that several lncRNAs display both oncogenic or tumor suppressive functions in a cancer-specific fashion. For example, *H19* has been shown to possess oncogenic properties in multiple cancers such as colorectal cancer, breast cancer and hepatocellular carcinoma, whereas in human rhabdoid cancers, *H19* displays tumour-suppressive roles [57,58]. 

Similar observations have been reported for another lncRNA, *NEAT1,* which is involved in paraspeckle formation. Although it is mostly reported to possess oncogenic roles in multiple cancers such as glioma, ovarian cancer, melanoma and prostate cancer, its loss has been shown to promote tumorigenesis in a mouse model of pancreatic ductal adenocarcinoma [59,60,61]. Another lncRNA that displays context-specific function as an oncogene or tumor suppressor is *CASC15*, which has been shown to act as an oncogene in skin cancers whereas as a tumor suppressor in the context of Neuroblastoma [36,62]. *TINCR* (terminal differentiation-induced non-coding RNA) is another example of lncRNAs with dual functions in the progression of cancer. While it adopts a tumor suppressive role in glioma, prostate cancer and retinoblastoma; it works as an oncogene in gastric, bladder and breast cancer [21]. 

These examples indicate towards dynamic nature of lncRNA molecules which can adopt contrasting roles in a cell of the origin-specific manner in tumour development. Additionally, the tumor microenvironment could also influence the functional role of a given lncRNA in a genetic context. Do such lncRNAs have special structural or regulatory features that provide them the adaptability in a given tumor context remains unknown? Finally, treatment approaches that could alter the expression of such lncRNAs should be used with caution.

### 3.4. LncRNAs Influence Tumorigenesis through Metabolic Pathways

Metabolic reprogramming is one of the key hallmarks associated with cancers. In addition to protein coding genes, lncRNAs are also reported to participate in regulation of metabolic pathways. For example, lncRNA *SAMSON* promotes melanoma survival by enhancing mitochondrial function [63]. It regulates the maturation of mitochondrial 16S rRNA via interaction with p32. Overexpression of *SAMSON* creates a unique metabolic dependency in melanoma cells; targeting the lncRNA leads to reduced growth/viability of invasive melanoma cells. Another lncRNA, Tp53-regulated inhibitor of necrosis (*TRINGS*) gets elevated after glucose deprivation in cancer cells. *TRINGS* promotes tumour growth (in vitro and in vivo) via binding to STRAP and inhibits the STRAP-GSK3β-NF-κB necrotic pathway [64]. Other lncRNAs such as *lincRNA-p21* and FoxO-induced long non-coding RNA 1 (*FILNC1*) are also involved in regulating glycolysis [65,66]. HIF-1α induced *lincRNA-p21* disrupts VHL-HIF-1α interaction by binding to both the molecules and leads to a collection of HIF-1α, and thus hypoxia-enhanced glycolysis. On the other hand, *FILNC1* is a translation suppressor of c-Myc RNA, and thus, its downregulation enhances the Warburg effect through c-Myc upregulation. Contrary to *FILNC1*, using siRNA screen, Sang et al. have identified a lncRNA for calcium-dependent kinase activity (CamK-A) that promotes the Warburg effect. This lncRNA is highly expressed in various types of cancer and is involved in remodelling the tumour microenvironment via activation of calcium trigged signalling [67]. LncRNAs can also act as signalling molecules that transmit signals between immune and tumour cells to promote the Warburg effect. For instance, Chen et al. have reported that tumor-associated macrophages (TAM) enhances aerobic glycolysis via synthesis and secretion of an extracellular vesicle packaged HIF-1α-stabilizing long non-coding RNA (*HISLA*) [68]. Taken together, lncRNAs play crucial roles in regulating metabolism and consequently impact the associated process of the tumor microenvironment. 

### 3.5. LncRNAs Control Tumorigenesis through Regulating Immune Response and Tumour Microenvironment

Many lncRNAs have been reported with regulatory roles in various aspects of immune regulation, including hematopoietic development, myeloid differentiation, CD4+ T cell differentiation, inflammatory response, and T cell activation, etc. [69]. However, whether these non-coding RNA molecules have any roles in antitumor response. What type of lncRNAs are those, and how do they perform their functions? Do lncRNAs influence the tumour microenvironment? Most importantly, as immunotherapy is at the forefront of cancer treatment, can these non-coding transcripts improve clinical outcomes in patients receiving cancer immunotherapy. At large, these pertinent questions mostly remain unanswered. However, recent reports have shed some light on this unexplored area. By comparing TCGA data of immunologically ‘hot’ with ‘cold’ human melanoma tumours, Li et al. have identified an immunogenic lncRNA, LIMIT (lncRNA inducing MHC-I and immunogenicity of tumour), with immunoregulatory roles [70]. It is a cis-acting lncRNA that gets stimulated by interferon-γ (IFNγ) and induces upregulation of major histocompatibility complex 1 (MHC-I) expression on tumour cells but does not affect PD-L1. Mechanistically, *LIMIT* regulates the expression of MHC-1 via the *LIMIT*– guanylate-binding protein (GBP)–heat shock factor (HSF1) regulatory axis. Loss of IFNγ and MHC-I signatures are frequently observed in human tumours that can cause unresponsiveness to immunotherapy. By employing CRISPR-mediated upregulation of *LIMIT*, the authors demonstrated that the lncRNA dramatically drove MHC-1 expression and sensitised tumour cells to T cell-mediated response against the tumour. In contrast, lncRNAs such as *LINK-A* (long intergenic non-coding RNA for kinase activation) and *MALAT1* have been reported as inhibitors of T cell activation. In breast cancer, *LINK-A* promotes degradation of antigen peptide-loading complex (PLC), required to present tumour antigens for activating T cells [71]. In diffused large B-cell lymphoma (DLBCC), *MALAT-1* induces the expression of programmed death-ligand 1 (PD-L1) via sponging microRNA-195 (miR-195), resulting in inhibitory signals for T cell activation [72]. The screening population for both these RNAs can identify non-responders. Additionally, CRISPR-mediated inhibition needs to be tested if inhibiting these RNAs sensitises the T cells to immunotherapy. 

Recent studies have also implicated lncRNAs as regulators of inflammatory response against microbial infections. For example, *lincRNA-Cox2* and *Inc-DC* interact with transcription factors such as NF-kB and STAT3, respectively, to regulate dendritic cells (DC) and macrophages [73,74]. Furthermore, *lincRNA-cox2* also interacts with hnRNP A/B and A2/B1 and modulates the inflammatory response via regulating the expression of IFN-stimulated genes and chemokines.

Evidence of lncRNAs influencing the tumor microenvironment (TME) is also reported. TME consists of nonmalignant stromal cells such as tumour-associated macrophages (TAMs) and cancer-associated fibroblasts (CAFs). Chen et al. identified a lncRNA, *LNMAT1* (Lymph Node Metastasis Associated Transcript 1), that epigenetically activates CCL2 expression, leading to TAM infiltration and promoting lymph node metastasis via VEGF-C excretion in bladder cancer [75]. 

Interestingly, a recent study by Li et al. has characterised lncRNAs as regulators of immune pathways across 33 cancer types. The researchers have developed a comprehensive and interactive database, ImmLnc that can be used to query the function of any lncRNA in immune regulation. This pan-cancer study will aid in further understanding the role of lncRNA in immune modulation during cancer progression [76]. 

## 4. Current and Emerging Approaches for Functional Characterization of LncRNAs

Despite dysregulation in lncRNA expression in many tumor types, there seems a paucity of functionally characterized transcripts. With advances in techniques for genetic manipulation, it has become possible to characterise lncRNAs in bulk or individually. Additionally, among the functionally known candidates, it is crucial is to establish whether the transcript itself or the DNA element that encodes the transcript is responsible for the observed phenotype. For instance, Cho et al. have shown that the promoter of lncRNA *PVT1* regulates neighbouring MYC gene expression rather than the transcript [77]. In the current section, we will discuss how genetic approaches in cell and animal models have added to our functional understanding of a number of lncRNAs (Figure 2). Further, we will also discuss computational approaches/databases available for functional characterization of lncRNAs.

### 4.1. CRISPR-Based Genetic Screens as Discovery Platforms for LncRNA Functions

Genetic screens have been widely used to identify protein-coding genes essential for survival of the cancer cells (depletion screens) and the genes whose loss stimulate cell proliferation (enrichment screens) [78,79,80]. Additionally, they are also used to screen for genes whose presence or absence is associated with sensitivity of cancer cell to a pharmacological inhibitor (drug screens). Compared to applying CRISPR-based screening technologies to protein-coding genes, fewer studies have applied these genetic approaches to lncRNAs. For instance, only three screens have been reported so far targeting lncRNAs in cell growth. Zhu et al. constituted a lentiviral paired sgRNA library (pgRNA) targeting 700 human lncRNAs with known or alleged roles in cancers [81]. The genetic screen performed on HuH7.5 hepatocellular carcinoma cells revealed four oncogenic RNAs (*LINC00176*, *LINC01087*, *LINC00882,* and *LINC00883*) that promote cell proliferation and five tumor-suppressor lncRNAs that negatively regulate cell proliferation (*AC004463.6*, *AC095067.1*, *HM13-AS1*, *RP11-128M1.1,* and *RP11-439K3.1*). Such approaches come with a caveat that it is difficult to establish whether the transcript itself or the DNA element that encodes the transcript is responsible for the observed phenotype. For example, Cho et al. have shown that the promoter of lncRNA *PVT1* regulates neighbouring MYC gene expression rather than the transcript [77]. To circumvent this limitation, Liu et al. have used a CRISPR-mediated interference (CRISPRi) approach, where dCas9 is fused with the repressive KRAB domain, resulting in sgRNA mediated target gene suppression [82]. They used CRISPR library for lncRNAs (CRiNCL) targeting 16, 401 loci (10sgRNA per TSS) in seven cancer cell lines and one iPSCs. They found role of 499 lncRNAs in cell growth, however, most of these candidates were specific to one cell type only. These observations underscore the cell-type and context-specific roles of lncRNAs and underline the specificity and sensitivity of CRISPR based screening methods. In addition to these two screens, Liu et al. have constructed a CRISPR library targeting splice sites of 10,966 RNAs and identified 230 lncRNAs with functional roles in cell growth of K562 cells [83]. Encouraging results for these screens showcase the potential of such approaches in unraveling lncRNA function in growth of cancer cells. 

In addition to applying the CRISPR/CAS9-based functional screens in identifying lncRNAs associated with tumor growth, these screens have been applied to identify lncRNAs in drug resistance/sensitivity. Using activator CRISPR (CRISPR-a) and sgRNA library, Joung et al. have identified an lncRNA, *EMICERI*, whose transcript or its transcription is required for positive regulation of *MOB3B* gene responsible for inducing vemurafenib resistance in melanoma cells [84]. Similarly, using CRISPRi library, Liu et al. have reported 33 lncRNAs that sensitize glioma cells to a clinically relevant dosage of radiation [85]. Particularly, antisense oligo (ASO) targeting of *lncGRS-1* transcript reduces tumour growth and sensitivity to radiation therapy in human brain organoids.

In summary, CRISPR-based approaches could serve as an efficient tool for the discovery of lncRNA function during various steps of cancer progression as well as treatment. Furthermore, as in the case of protein-coding genes, attempts are being made to perform lncRNA screens in vivo. For example, Attenello et al. have described the methodology for CRISPRi screen in vivo for glioblastoma cells and listed an option for screening lncRNA during tumour growth in a mouse model of brain cancer [86]. Liu et al. too have used CRSIPRi screen in vivo for investigating the functional roles of Wnt-regulated lncRNAs in pancreatic cancer. They identify 34 Wnt-regulated lncRNAs having a functional effect on cancer cell growth in xenograft model [21]. 

### 4.2. Animal Model Systems for Studying LncRNA Function

Animal cancer models provide means to look at various aspects of cancer development and progression and efficient systems for testing the efficacy of therapeutic strategies. As lncRNAs have gained relevance in several malignancies, similar to protein-coding genes, investigating their functional roles in animal models is necessary. However, unlike protein-coding genes, studying lncRNA function in animal models is more complex. First, not all lncRNAs are conserved between mouse and humans. Therefore, alternative approaches such as xenografts, organoid cultures, patient-derived xenografts (PDX), patient-derived tumour organoids (PDTO), or Zebrafish models must be adopted. Secondly, for conserved lncRNAs, models involving deletion of genomic regions need to be careful such that regulatory sequences that regulate the expression of the lncRNA or neighboring genes are not disturbed. Especially, these approaches should be able to distinguish between the act of transcription at the lncRNA locus vs the transcript per se and possibly attribute the phenotype to lncRNA. Thirdly, as the nature of lncRNAs is to fine-tune gene expression in a spatio-temporal manner, the absence or overexpression of RNA alone may not be able to trigger tumour formation in mice. In other words, they may not serve as initial drivers or initial lesions but may act as accelerators or inhibitors of tumour progression once the primary lesion has been established. Therefore, developing compound mouse models where lncRNA expression can be conditionally manipulated or regulated in a tissue-specific manner with a background of gain-of-function and loss-of-function known oncogenes and tumour suppressor genes, respectively. 

We will discuss a few examples of lncRNAs that have been studied in vivo to further substantiate on all the points mentioned above. *MALAT1* is one such lncRNA whose elevated levels are seen in numerous human cancers and is associated with poor prognosis in several human malignancies. Surprisingly, the mice lacking germline *MALAT1* (KO mouse) are normal suggesting the plausible redundancy in its regulatory functions. Although the absence of *MALAT1* is sustained by cells in normal development, its elevated levels in cancer settings may contribute to the deregulation of cancer-specific pathways. In fact, functional roles of *MALAT1* have been elucidated in mouse models of breast cancer, however, variable data has been seen between studies possibly due to differences in the approach adopted for knockout strategies. Arun G et al. removed the promoter of *MALAT1* transcript and demonstrated an oncogenic role of *MALAT1*, whereas studies by Kim et al. used the same background and reported tumour-suppressive functions for the same RNA when the transcript was truncated prematurely using transcription termination (PolyA+) signal [87,88]. These findings underscore the need to delineate the impact of locus control regions or the act of transcription or transcript per se while designing mouse models in lncRNA investigations. Future investigations with other KO strategies may shed more light on the role of *MALAT1* in vivo. Another elegant use of mouse genetics by *Mello et al.* has shown that the p53-dependent lncRNA *NEAT1* acts as a barrier to activated KrasG12D driven pancreatic ductal adenocarcinoma (PDAC) development, and its knockout leads to the formation of multiple preneoplastic lesions [61]. Further, a conditional KO approach has been adopted in case of lncRNA *IRENA*. Studies in this animal model suggest *IRENA* as a promising candidate for overcoming chemotherapy resistance, as its deletion IFN-activated macrophages abrogates their pro-tumour property [89]. Similarly, KO mouse model of cancer-testis lncRNA in hepatocellular carcinoma (*lnc-CTHCC*) revealed its pro-oncogenic properties via association with hnRNPK and YAP1 transcriptional activation [90]. 

In parallel to genetic studies in mouse, patient-derived xenograft models can be used for investigating the significance of lncRNA in tumor progression as well as revealing the therapeutic efficacy of lncRNA. In fact, orthotopic PDX models are clinically superior systems for investigating the role of lncRNA in cancer progression. Establishing PDX from multiple patients provides a stringent way of evaluating a therapeutic strategy in vivo, as all individuals may carry their unique passenger mutations and these efforts form basis for lncRNA-based personalized treatment procedures. Thus, it presents a scenario similar to what may be encountered in patients. For example, Lucci et al. have derived two PDX models from different metastatic melanoma lesions with BRAF (V600E) mutations. Using these models, the authors report that targeting lncRNA *SAMSON* sensitises melanomas to inhibitors of the MAP-kinase pathway [63]. Similarly, Fang et al. established 9 PDX models of non-small cell carcinoma (NSCLC), comprising both lung squamous cell carcinoma (LUSC) and lung adenocarcinoma (LUAD). Importantly, these models closely resemble the original tumour in terms of histo-pathology, immunohistochemical features, and expression profile. These observations underscore the utility of PDX models for lncRNA research specifically for those candidates that are expressed only in human beings. Using the NSCLC PDX models, the authors investigated the anti-tumour efficacy of silencing of *TUG1* and *LCAL6* lncRNAs using siRNAs [91]. It was found that *TUG1* inhibition suppressed tumour growth unlike *LCAL6* where no significant effect on tumour growth was seen. Likewise, the anti-tumour efficacy of silencing of S-phase specific lncRNA *SCAT7* was investigated on a stage IV metastatic LUAD PDX model using locked nucleic acid-modified antisense oligonucleotides (LNA-ASOs) [4]. Silencing of *TUG1* and *SCAT7* significantly reduced tumour growth. 

Finally, three-dimensional (3D) organoid cultures are attractive methods of characterizing the therapeutic efficacy of lncRNA. Luo et al. have used three-dimensional (3D) patient-derived breast cancer organoids and reported the therapeutic potential of targeting Aurora A/PLK1–associated long non-coding RNAs (APAL) in breast cancer [92].

The above discussed examples clearly underline the importance of genetic approaches in cell and animal models of cancer. Hopefully, future studies will add to our current understanding about lncRNA functions using these functional approaches.

### 4.3. Computational Approaches and Functional LncRNAs

In conjunction to genetic approaches and animal models, computational tools can also significantly contribute towards functional characterization of lncRNAs. There are dedicated databases which can provide significant information towards functional understanding of a lncRNA in a given condition. For example, Lnc2Cancer 2.0 is a database for exploring the relationship between lncRNAs and different cancer subtypes [93]. Its latest version includes circular RNAs (ciRNAs) and has included data from latest RNA-sequencing (RNA-seq) and single cell RNA-seq studies [94]. The lncRNA cancer score can be utilized in predicting the functional relevance of a known or unknown lncRNA in a given cancer or its specific subtype. Co-lncRNA is another interesting tool that exploits the co-expression patterns of mRNAs with lncRNAs. It can be used to predict gene ontology annotations and KEGG pathways associated with a given lncRNA–mRNA co-expression network [95]. For instance, Wu et al. have identified distinct lncRNA–mRNA co-expression patterns between normal and tumour breast tissue [96]. The analysis yielded a candidate lncRNA, *AC145110.1,* which is differentially expressed with mRNAs enriched in biological functions such as cell proliferation and cell to cell signaling. Moreover, *AC145110.1* was highly correlated with clinical outcomes in breast cancer. Recently, Vancura et al. have developed an excellent database for cancer associated lncRNAs named as Cancer LncRNA Census 2 (CLC2). By integrating the results of functional studies such as mutagenesis screens, genetic screens along with features such as differential expression, survival outcomes, copy number variations, they have curated a list of 492 RNAs, whose functional attributes in cancer can be predicted with high confidence [97]. 

## 5. LncRNA Based Therapies and Diagnostic Markers

Due to the well-established roles of lncRNAs in cancer pathogenesis, various strategies are under development targeting these molecules, mainly siRNA-based drugs, antisense oligonucleotides (ASOs) and inhibitors against lncRNAs [98]. The therapeutic potential of siRNAs has been evaluated in preclinical models; for example, siRNA cloned in lentiviral vector has been used to target *CASC9* lncRNA in esophageal squamous cell carcinoma resulting in reduced tumour invasion and migration [99]. Similarly, targeting oncogenic lncRNA, *DANCR*, with siRNA nanoparticles showed tumour suppression in a mouse model of triple-negative breast cancer (TNBC) [100]. In comparison to siRNAs, ASOs are more efficient in targeting lncRNAs. Targeting overexpressed lncRNA, *MALAT1,* in a mouse model of breast and xenograft models of lung cancers using ASOs significantly reduced tumour growth and metastasis [87,101]. Another study by Liu et al. used ASO against lncUSMycN and observed a reduction in tumour size in the neuroblastoma xenograft model [102]. In addition, modified ASO such as LNAs and cEt modifications are under development, making ASO as one of the most promising approaches for targeting lncRNAs. 

In addition to siRNAs and ASO, small molecule inhibitors targeting lncRNA-protein interactions are also reported. Fatemi et al. developed an alpha screening technology to screen molecules that destabilise RNA-protein interactions. They identified a small molecule, ellipticine, as an inhibitor of lncRNA *BDNF-AS* and *HOTAIR’s* interaction with the epigenetic repressive modifier EZH2 [103]. Furthermore, Abulwerdi et al. employed a microarray strategy to screen inhibitors targeting triple helical structures at the 3’ end of *MALAT1* RNA. Two molecules identified from the screen could reduce *MALAT1* levels and branching morphogenesis in an organoid model of breast cancer [104]. In Ewing sarcoma’s, too, a small molecule YK-4—279 is reported as an inhibitor of HULC lncRNA’s regulatory network [105]. Although the studies with small molecule inhibitors of lncRNAs are limited, these molecules are cost-effective and easy to administer compared to siRNAs or ASOs. Therefore, more lncRNA inhibitors must be screened and validated in cell and/or animal cancer models for their efficacy and safety.

In addition to above discussed lncRNA associated therapies, these molecules have the potential to be used as diagnostic markers. This is best elucidated in case of *PCA3* which is the first and only lncRNA that has been approved by the Federal Drug Administration (FDA) as a biomarker for prostate cancer [106,107]. It can be easily detected in the urine samples of prostate cancer patients.

## 6. LncRNA: Reality or Hype-a Future Perspective

In recent years, several studies have implicated lncRNA in various aspects of oncology research. These non-coding molecules have roles in cancer development, progression, diagnosis, and treatment landscape. Despite significant progress in our understanding of lncRNA role in tumor development and progression in the past few years, only *PCA3* has been approved by FDA for its use in the clinics for diagnosis. Therefore, after more than decade of research in this field of lncRNA, it is essential to reflect whether these RNA molecules have the potential to improve patient outcomes or there remain issues that need to be addressed before lncRNAs can be actively used in the clinic.

Among many roadblocks in the lncRNA research, the following issues need to be primarily addressed:Most studies have confined themselves in correlating expression changes in lncRNAs with either risk status of cancer or cancer subtypes.Barring a few exceptions, there is a lack of studies exploring or validating the role of lncRNAs in cancer mouse models/ PDX- models.Lack of models of lncRNAs expressed only in humans.The mechanism behind lncRNAs action in specific cancer contexts remain poorly understood, including their downstream effectors or upstream regulators.Distinguishing between roles of lncRNAs as drivers or passengers in cancer progression.Lack of inhibitors and studies with drugs against either lncRNAs or lncRNA-protein interactions.Lack of in vivo genetic screens.

Despite these challenges associated with lncRNAs, these molecules hold much promise for future cancer research. Some of the outstanding features which support lncRNA as a new candidate in cancer research are as follows:Spatio-temporal expression of lncRNAs make them ideal candidates for biomarkers.Recent bio-informatic studies have elucidated that lncRNAs can act as a predictor of therapy response.Mechanistic and in vivo studies in animal models have elucidated the functional role of lncRNAs in cancer progression.LncRNAs have great potential as drug targets due to their roles as regulators of vital biological processes.

Taken together, studies with lncRNAs in cancer have opened up a new and exciting avenue for understanding cancer pathogenesis and their potential therapeutical significance. More robust and loss of function approaches in relevant cancer models will be vital in extracting the true clinical potential of these promising non-coding species. 

## 7. Conclusions and Future Perspective

In recent years, lncRNAs have gained a lot of traction due to their broad range of functions in multiple cellular and developmental processes. Moreover, their expression levels have been found to be significantly associated with clinical outcomes in almost all the human cancers. These observations have generated a lot of excitement in the scientific community. The fundamental question that needs to be addressed is if lncRNAs have decisive roles in the initiation of cancer. Although their impact on the aggressiveness of cancer has been seen in some cases, the field in general suffers from the lack of enough functional studies. In the current review, we have compiled the examples of lncRNAs that have been well studied for their roles in the process of carcinogenesis. Furthermore, we lay out a road map of approaches that can be used for functional characterisation of lncRNAs in the development of cancer. The diversity and tissue-specific role of lncRNAs make them attractive molecules for developing focused treatment modalities as well as a precise diagnostic marker. Hopefully, future functional studies will enhance our current understanding of lncRNAs and thereby accelerate the development of lncRNA-based diagnostics and therapeutics.

## Figures and Tables

**Figure 1 cancers-14-04760-f001:**
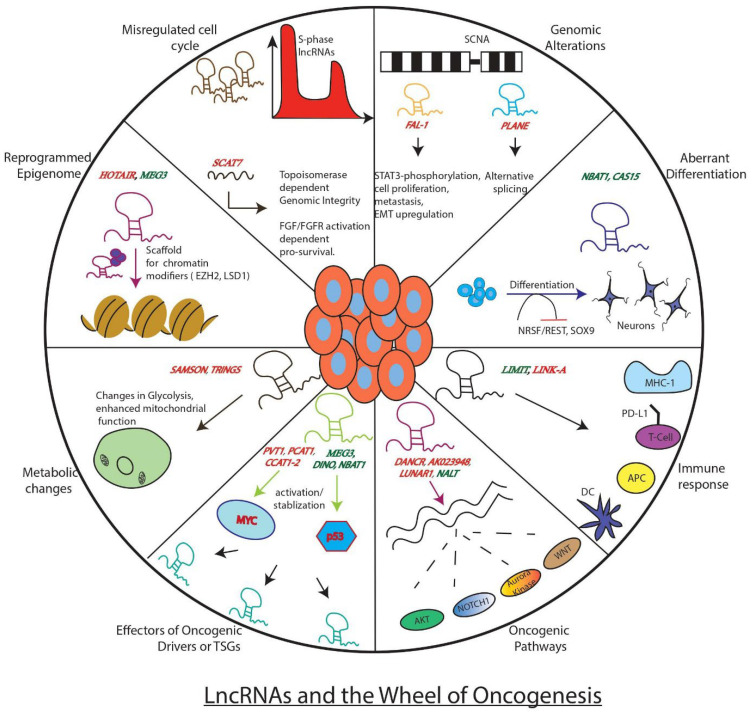
Well known examples of LncRNAs and how they influence key genes, processes, or pathways involved in tumor development. The figure depicts the regulatory role of lncRNAs as either promoters of the wheel of oncogenesis (red) or as brakes (green) inhibiting its movement.

**Figure 2 cancers-14-04760-f002:**
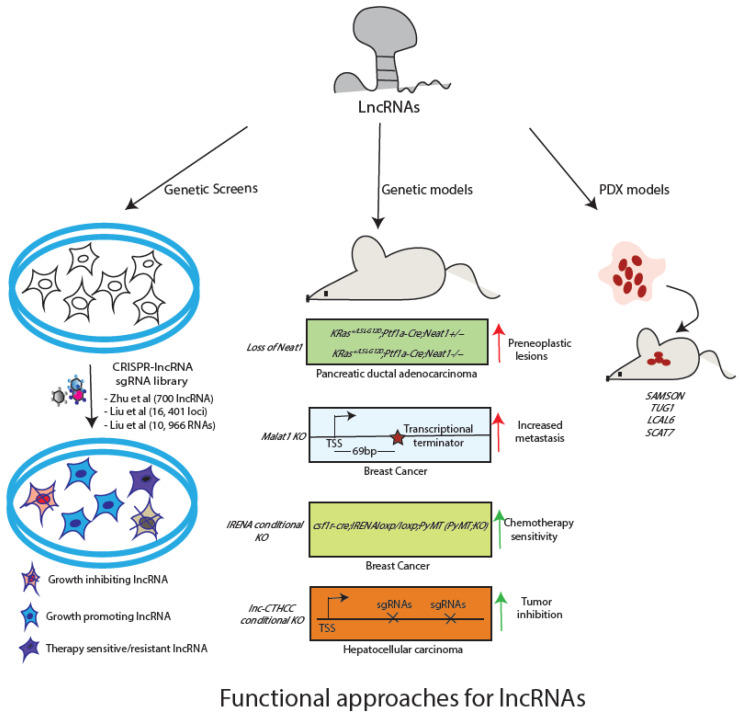
Available approaches for functional characterization of lncRNA. Genetic screens can be used for identification of functional lncRNAs. Individual candidates can be studied using mouse models or PDX models.

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
