# Peer review of "Long Non-Coding RNAs: Tools for Understanding and Targeting Cancer Pathways"

_cancers, 2022, doi:10.3390/cancers14194760_

Round 1

Reviewer 1 Report

Pandey and Kandri

Long noncoding RNAs: Tool for understanding and targeting cancer pathways

The manuscript needs some English editing specially after point 3: line 249.

Sentences between lines 167-172 are not clear.

A first section should be added: long-noncoding RNA structure, location in the genome and roles.

Some abbreviations need to be defined when used for the first time. For example, GWAS, SNPs, etc

The sections and subsections should be better divided. For example, section 2: 2.1, 2.2, etc.)

Some references are missing.

Sec 2.3 (Context-specific roles of lncRNAs as oncogenes or TSGs) is to short and should be explained in more detail.

The section of Functional approaches for lncRNA needs to be expanded: computational approaches are very important, also, techniques as SAGE, CAGE, ect should be mentioned. Also FISH to better identify the location of certain lncRNAs.

The authors mentioned (lines 330-332) a study of Li et al. The results of this study should be expanded.

Lines 355 & 356 established five oncogenic RNAs that promote cell proliferation and four tumor suppressor lncRNAs that negatively regulate cell proliferation. Can the author mention which five and four respectively?

The author mentioned in line 462 that PCA3 is the only lncRNA that FDA has approved for its use in clinics for diagnosis. This is a significant fact; however, the author fails to mention or describe PCA3 in the manuscript. I suggest a section on lncRNAs currently in use at clinical levels where PCA3 can be fully described.

The section 4: LncRNA: reality…should be completely re-written.

Better: Conclusions and Future directions.

Author Response

Reviewer 1:

Long noncoding RNAs: Tool for understanding and targeting cancer pathways

We thank the reviewer for his/her suggestions. Please see our point wise response to each comment.

The manuscript needs some English editing specially after point 3: line 249.

The manuscript has been edited for English, especially after point3.

Sentences between lines 167-172 are not clear.

The sentences have been modified.

A first section should be added: long-noncoding RNA structure, location in the genome and roles.

A new section has been added.

Some abbreviations need to be defined when used for the first time. For example, GWAS, SNPs, etc

Abbreviations have been defined.

The sections and subsections should be better divided. For example, section 2: 2.1, 2.2, etc.)

Sections and sub-sections have been divided.

Some references are missing.

New references have been added.

Sec 2.3 (Context-specific roles of lncRNAs as oncogenes or TSGs) is to short and should be explained in more detail.

The section has been explained.

The section of Functional approaches for lncRNA needs to be expanded: computational approaches are very important, also, techniques as SAGE, CAGE, ect should be mentioned. Also FISH to better identify the location of certain lncRNAs.

A new section has been added regarding computational approaches.

The authors mentioned (lines 330-332) a study of Li et al. The results of this study should be expanded.

The results of Li et al. have been expanded.

Lines 355 & 356 established five oncogenic RNAs that promote cell proliferation and four tumor suppressor lncRNAs that negatively regulate cell proliferation. Can the author mention which five and four respectively?

The names of the lncRNAs have been added.

The author mentioned in line 462 that PCA3 is the only lncRNA that FDA has approved for its use in clinics for diagnosis. This is a significant fact; however, the author fails to mention or describe PCA3 in the manuscript. I suggest a section on lncRNAs currently in use at clinical levels where PCA3 can be fully described.

PCA3 has been discussed in the review.

The section 4: LncRNA: reality…should be completely re-written.

A conclusion and future direction section has been added to support this section.

Better: Conclusions and Future directions.

This section has been added.

Reviewer 2 Report

Pandei and Kanduri resume current knowledge concerning the important questions if and how lncRNAs play critical roles in the etiology and progression of cancer. They first provide an overview of different lncRNA functional classes based on a variety of known regulatory transcripts, and then resume in vitro and in vivo approaches to determining the functions of lncRNAs and their potential roles in cancer before they end with key challenges and accomplishments in the field. 

The text is very well written, easy to read and the illustrations are clear. Perhaps the authors should cite recent comparable reviews (such as Rinn and Chang, Annu Rev Biochem. 2020, Nandwani et al., Cancer Lett 2021), and explain briefly in the introduction what distinguishes their manuscript from such published articles. 

Moreover, the authors ought to include the excellent work carried out by the Johnson lab on the Cancer lncRNA Census (CLC) project (Vancura et al., Nucleic Acids Res 2021). This is an omission in an article that claims to resume current functional knowledge about lncRNAs in cancer. 

Another issue concerns work done in the field of translational research that the authors appear to identify as lacking (line 479). I suggest adding a section that covers studies of therapeutical antisense oligonucleotides (summarized by Winkle et al. Nat Rev Drug Discov. 2021). 

Likewise, the authors claim that there is a lack of in vivo genetic screens for functional lncRNAs (line 479). Rather than that, I recommend including a paragraph about relevant work reported by, among others, Ruan et al. Nat Communications 2020, Atenello et al. J Neurosci Res 2021 (which the authors actually mention in their review), and Liu et al. Genome Med 2020. 

Author Response

Pandei and Kanduri resume current knowledge concerning the important questions if and how lncRNAs play critical roles in the etiology and progression of cancer. They first provide an overview of different lncRNA functional classes based on a variety of known regulatory transcripts, and then resume in vitro and in vivo approaches to determining the functions of lncRNAs and their potential roles in cancer before they end with key challenges and accomplishments in the field. 

 We thank the reviewer for his/her suggestions. Please see our point wise response to each comment.

The text is very well written, easy to read and the illustrations are clear. Perhaps the authors should cite recent comparable reviews (such as Rinn and Chang, Annu Rev Biochem. 2020, Nandwani et al., Cancer Lett 2021), and explain briefly in the introduction what distinguishes their manuscript from such published articles. 

The reviews have been cited and introduction has been modified.

Moreover, the authors ought to include the excellent work carried out by the Johnson lab on the Cancer lncRNA Census (CLC) project (Vancura et al., Nucleic Acids Res 2021). This is an omission in an article that claims to resume current functional knowledge about lncRNAs in cancer. 

A new section about computational approaches has been added where above mentioned paper has been discussed.

Another issue concerns work done in the field of translational research that the authors appear to identify as lacking (line 479). I suggest adding a section that covers studies of therapeutical antisense oligonucleotides (summarized by Winkle et al. Nat Rev Drug Discov. 2021). 

A new section about therapeutics and diagnostic marker has been added.

Likewise, the authors claim that there is a lack of in vivo genetic screens for functional lncRNAs (line 479). Rather than that, I recommend including a paragraph about relevant work reported by, among others, Ruan et al. Nat Communications 2020, Atenello et al. J Neurosci Res 2021 (which the authors actually mention in their review), and Liu et al. Genome Med 2020. 

The suggested papers have been added and discussed in the manuscript.

Reviewer 3 Report

The Review submitted by Kanduri and Pandey summarized an important issues of the role of lncRNAs in oncogenesis and discussed possibilities and drawbacks of different approaches for functional characterization of lncRNA in cancer. The paper is clearly written, and well-organized. I only have minor issues:

1.      Figure 1 could better reflect section in the text. In the present form, Figure 1 rather confused me that clarified and organized functions of lncRNAs. The authors in the legend of Figure 1 state “The figure depicts the regulatory role of lncRNAs as either promoters of the wheel of oncogenesis or as brakes inhibiting its movement”. It would be beneficial to indicate in the figure which lncRNAs are promoters, brakes or both depending on context. Additionally, some statements in the figure are misleading e.g. it the figure it seems as PLANE induce splicing and not regulate alternative splicing.

2.      Figure 1 also lacks important lncRNAs such as NORAD, NEAT and PCA3.

3.      Legends of both figure 1 and figure 2 should explain better the idea and content of the figures.

4.      Page 4, line 103, In the sentence “For example, cell cycle, one of the well-studied cancer hallmark…” it should rather “… deregulated (or misregulated) cell cycle…

5.      Figure 2, In comparison to genetic models, the PDX models seems to be presented in a very brief way. It would be beneficial to include the cancer type and outcome of the model also for PDX models.

6.      Page 12, line 462-463: The authors state “…PCA3 has been approved by FDA for its use in the clinics for diagnosis” and that is only sentence concerning PCA3. In my opinion, the authors should also include more information about function and application of lncRNA PCA3, at least as a reference for another review paper and a couple sentence summary. 

Author Response

The Review submitted by Kanduri and Pandey summarized an important issues of the role of lncRNAs in oncogenesis and discussed possibilities and drawbacks of different approaches for functional characterization of lncRNA in cancer. The paper is clearly written, and well-organized. I only have minor issues:

We thank the reviewer for his/her suggestions. Please see our point wise response to each comment.

  1. Figure 1 could better reflect section in the text. In the present form, Figure 1 rather confused me that clarified and organized functions of lncRNAs. The authors in the legend of Figure 1 state “The figure depicts the regulatory role of lncRNAs as either promoters of the wheel of oncogenesis or as brakes inhibiting its movement”. It would be beneficial to indicate in the figure which lncRNAs are promoters, brakes or both depending on context. Additionally, some statements in the figure are misleading e.g. it the figure it seems as PLANE induce splicing and not regulate alternative splicing.

The figure has been modified as per reviewer’s suggestions.

  1. Figure 1 also lacks important lncRNAs such as NORAD, NEAT and PCA3.

These lncRNAs have been discussed in the text and is technically difficult to add these examples as well.

  1. Legends of both figure 1 and figure 2 should explain better the idea and content of the figures.

The legends have been described.

  1. Page 4, line 103, In the sentence “For example, cell cycle, one of the well-studied cancer hallmark…” it should rather “… deregulated (or misregulated) cell cycle…

The line has been modified.    

  1. Figure 2, In comparison to genetic models, the PDX models seems to be presented in a very brief way. It would be beneficial to include the cancer type and outcome of the model also for PDX models.

The PDX models have been elaborated.

  1. Page 12, line 462-463: The authors state “…PCA3 has been approved by FDA for its use in the clinics for diagnosis” and that is only sentence concerning PCA3. In my opinion, the authors should also include more information about function and application of lncRNA PCA3, at least as a reference for another review paper and a couple sentence summary. 

PCA3 has been discussed in the manuscript.

Round 2

Reviewer 1 Report

Still there are some grammar errors. Example:

Lines 304-305: "These examples indicate towards dynamic nature of lncRNA molecules which can 304 adopt contrasting roles in a cell of the origin-specific manner in tumour development". Revise this sentence.

Lines 443-444: They identify 34 Wnt-regulated lncRNAs having functional 443 effect on cancer cell growth in xenograft model[21]. past tense: identified.

Line 446: Animal cancer models... I think that: Cancer animal models or animal models for cancer research, is more used.